# Combination treatment of dendrosomal nanocurcumin and low-level laser therapy develops proliferation and migration of mouse embryonic fibroblasts and alter TGF-β, VEGF, TNF-α and IL-6 expressions involved in wound healing process

**Afsaneh Ebrahiminaseri[1], Majid Sadeghizadeh[1]\*, Ahmad Moshaii[2], Golareh Asgaritarghi[1], Zohreh Safari[1]**

1 Department of Genetics, Faculty of Biological Sciences, Tarbiat Modares University, Tehran, Iran,
2 Department of Physics, Tarbiat Modares University, Tehran, Iran

\* sadeghma@modares.ac.ir

## Abstract

### Introduction

Pressure ulcer (PU) is known as the third most costly disorder usually caused by prolonged pressure and stagnation in various parts of the body. Although several therapeutic approaches are employing, obstacles in appropriate healing for skin lesions still exist which necessitates new practical alternative or adjunctive treatments. Low level laser therapy (LLLT) as one of the mentioned new strategies have gained attention. Besides, curcumin is an herbal medicine extracted from turmeric with anti-inflammatory and antioxidative properties with promising beneficial therapeutic effects in wound healing. Employing dendrosomal nanoparticles, we overcome the hydrophobicity of curcumin in the present study. We hypothesized that combination treatment of DNC+LLLT (450 nm) simultaneously may promote the wound healing process.

### Material and methods

MTT assay, PI staining followed by flowcytometry, scratch assay and intracellular ROS measurement were used to investigate the effects caused by DNC and LLLT (450 nm) alone and in combination, on proliferation, cell cycle, migration and oxidative stress mouse embryonic fibroblast cells, respectively. The levels of growth factors and pro-inflammatory cytokines were evaluated by qRT-PCR and ELISA.

### Results

Our results indicated that combination exposure with DNC and LLLT leads to increased proliferation and migration of MEFs as well as being more efficient in significantly upregulating growth factors (TGF-β, VEGF) and decline in inflammatory cytokines (TNF-α, IL-6).

**Data Availability Statement:** All relevant data are within the paper and its Supporting information files.

**Funding:** We would like to acknowledge the Iran National Science Foundation (INSF) for its financial support of the project.

**Competing interests:** The authors have declared that no competing interests exist.

**Abbreviations:** DCFH-DA, 2',7'-dichlorodihydrofluorescein diacetate; DMEM, Dulbecco's modified Eagle's medium; DNC, Dendrosomal Nano Curcumin; ELISA, Enzyme-linked immunosorbent assay analysis; FBS, Fetal bovine serum; IL-6, Interleukin-6; j/cm$^2$, Jules/square centimeter; LLLT, Low-level laser therapy; MEFs, Mouse embryonic fibroblasts; MTT, 3-[4,5-dimethylthiazol-2-yl]-2,5-diphenyltetrazolium bromide; PBS, Phosphate-Buffered Saline; PU, Pressure ulcers; ROS, Reactive oxygen species; TGF-β, Transforming growth factor beta; TNF-α, Tumor necrosis factor alpha; UV–Vis, Ultraviolet–visible spectroscopy; VEGF, Vascular endothelial growth factor.

Moreover, findings of this research provide persuasive support for the notion that DNC could reduce the LLLT-induced enhancement in intracellular ROS in mouse embryonic fibroblasts.

## Conclusion

Concurrent exposure to anti-oxidant concentrations of DNC and LLLT enriched S phase entry and therefor increased proliferation as well as migration on MEFs through regulating the expression levels growth factors and shortening the inflammatory phase by modulating of cytokines. It should be noted that DNC were able to reduce the laser-induced oxidative stress, during wound healing, representing an informative accompaniment with LLLT.

## Introduction

Damage to the dermis, epidermis, muscle, and even the bone at various parts of the body especially in the bony areas caused by prolonged pressure and tissue ischaemia is referred as bedsores or pressure ulcers (PU). Due to deprivation of oxygen supply and nutrition aforementioned PU terminates in tissue necrosis and sometimes bring about fatal consequences by local and systemic infections [1–3]. PUs are frequent complications in long-term care known as the third most costly disorder following cancer and cardiovascular diseases [4]. Moreover, proper inhibitory interventions in order to decrease the prevalence of avoidable PUs have been reported [5]. According to a systematic review and meta-analysis study, global prevalence of PU in adult patients was reported around 12.8% and the incidence rate was 5.4 per 10,000 patient-day [6]. Bedsores have imposed high cost on patients and health services in countries every year (in the United States about $26.8 billion was estimated) [7].

Common treatments for PU including changing the patient's position, taking antibiotics, using ointments and even debridement and surgeries are costly and laborious representing a need for effective wound treatments without complexities [8, 9]. Novel treatments such as cell therapy, gene therapy, therapeutic ultrasound and electromagnetic therapies have been introduced as an alternative or adjunctive treatment in wound healing; but not fully responsive to healing especially in higher grade wounds. Recently, applications of photomedicine utilizing low level laser therapy (LLLT) as a nonsurgical technique have gained great attention for accelerating the wound healing procedure [10]. It has been shown that LLLT may stimulate cell proliferation, maintaining DNA integrity and the repair of damaged DNA whilst as low energy laser has negligible thermal effects also [10–12]. Blue light (wavelength range from 400 nm to 480 nm) which is part of the visible light spectrum is UV-free irradiation having fewer harmful side effects on mammalian cells compared with ultraviolet (UV) irradiation; since irradiation with blue light only reveals toxic effects at high dosages [13]. Blue light has attracted increasing attention in wound healing not only due to its intrinsic antimicrobial effect and inactivation of important wound pathogenic bacteria, including staphylococcus aureus and pseudomonas aeruginosa [14], but also owing to its favourable role in the treatment of inflammatory skin conditions [15] and the stimulation of wound closure [16].

Curcumin is a polyphenol extracted from the rhizome of Curcuma longa and well known as a multifunctional drug with anti-inflammatory, antioxidative, anti-infective and anti-carcinogenic activities owing to its varied influence on multiple signaling pathways [17, 18]. Forbye curcumin has a wider absorption range from 300–500 nm with maximum light absorption at 420 nm [13]. Besides having pharmacological safeties, mounting studies illustrated that this

herbal medicine has promising beneficial therapeutic properties in many diseases, including wound healing. However, the inherent limitations of the compound itself, such as hydrophobicity, instability, poor absorption and rapid systemic elimination, imposes a hurdle for translation to wider clinical applications [18, 19]. To overcome this major barrier and enhancing half-life and cellular permeability of curcumin, we employed an amphipathic and biodegradable nanocarrier which was previously synthesized by our group called dendrosomal nanocurcumin (DNC) [20–22].

The cutaneous wound healing process is generally considered to have four distinct phases: 1. haemostasis, 2. inflammation, 3. proliferation and 4. Remodeling. Numerous immune and non-immune cells, chemical agents and factors are involved in each phase [23]. Among cells, fibroblasts have been introduced as one of the key factors and have been investigated in the field of wound healing [24]. Additionally, the most important factors pointed out in several studies are transforming growth factor beta (TGF-β) as a growth factor involved in growth, survival, differentiation, migration, angiogenesis and as an anti-inflammatory cytokine [25] as well as vascular endothelial growth factor (VEGF) as the principal growth factor required in the angiogenesis process giving rise to vessel formation by supplying key nutrition supplements for the regenerated tissue [26]. Over-expression of these two factors from fibroblasts can help to improve the wound healing process [24]. On the other side, Tumor necrosis factor alpha (TNF-α) and Interleukin-6 (IL-6) are the most important pro-inflammatory cytokines in the process of wound healing released from haematopoietic and non- haematopoietic population such as fibroblasts. During the process of wound healing, TNF-α stimulates inflammation and growth factor production from immune cells primarily, the excess production of this cytokine provokes persistent inflammation yielding impaired wound healing as long as tissue damage [27]. Stimuli like persistent DNA damage and tissue destruction activates secretion of IL-6 by variety of cells including fibroblasts [28–30]. Over and above that, IL-6 expression is induced by TNF-α at the inflammatory stage as well [30]. Therefore, reducing inflammatory factors which are prolonged in chronic wounds may play a key role in shortening the inflammation phase, and also in promoting tissue repair and healing process by stimulating the proliferation stage [31, 32].

According to previous studies, LLLT and curcumin each their own were able to accelerate the wound healing process by stimulating growth factor production and therefore proliferation and migration of cells involved in repair process such as fibroblasts, reduction of inflammatory factors and shortening of the inflammatory phase, inducing angiogenesis, granulation tissue formation [33–37]. Also, it was declared that the progressive effect of these two therapeutic strategies was through regulating the production and accumulation of Reactive Oxygen Species (ROS) [38, 39]. The natural antioxidant curcumin can inhibit oxidative stress by scavenging free radicals like ROS or by inhibiting ROS-generating enzymes as well as activating ROS-neutralizing enzymes [39]. And LLLT, along with blue light, is suggested to increase levels of ROS and these alterations activate some transcription factors which lead growth factor production and cell proliferation [15, 40, 41]. Despite that, prolonged oxidative stress has been evident to mutilate wound healing process by inducing cell apoptosis. Thereby balance between the favorable act of ROS and their detrimental effects is demanded for proper wound repair [38].

On the basis of the concentration exerted, curcumin can exhibit antioxidant or pro-oxidant effects, therefor this study was performed to explore whether the antioxidant concentration of the compound (DNC) is able to induce proliferation and migration in MEFs cells in combination with LLLT (450 nm). And the alterations in growth factor (TGF-β, VEGF) and pro-inflammatory cytokine (TNF-α, IL-6) levels and measure of ROS accumulation was evaluated in mouse embryonic fibroblasts, after utilizing DNC and LLLT alongside each other.

## Material and methods

### Dendrosomal Nano Curcumin preparation (DNC)

Nano curcumin was purchased from Alborz Nano-drug Technology Company from Iran. For in vitro experiments, DNC was diluted in complete culture medium (DMEM) at different concentrations (0.25–10 μM) using an optimized protocol which has been set up in our lab [20–22, 42]. In Brief, different weight/weight ratios of dendrosome/curcumin were analyzed leading to the establishment of an appropriate ratio of 25:1. The loading of curcumin onto DNC was performed in which curcumin and dendrosome were co-dissolved in 5 mL of acetone followed by adding 5 mL of PBS, while stirring constantly. Acetone was evaporated by a rotary evaporator. The curcumin/dendrosome micelle solution was sterilized using a 0.22 μ syringe filter (Millex- LG, Millipore Co., USA). At the end, the prepared DNC was stored at 4 C in a light protected condition until use.

## DNC characterization

### Dynamic light-scattering analysis

Dynamic light scattering (DLS) (Zetasizer Nano ZS90; Malvern Instruments, Malvern, UK) was carried out to measure particle size distribution, polydispersity index (PDI) and zeta-potential of the DNC, using an argon laser beam at 633 nm and 90˚ scattering angle. (S1 Fig)

### Transmission electron microscopy

TEM images of DNC were taken by a Zeiss EM10C transmission electron microscope (Carl Zeiss AG, Oberkochen, Germany) photographed at an accelerating voltage of 100kV. In brief, the sample was diluted in distilled water (1mg/ml), subsequently a drop (10 μL) of DNC dispersion was placed on a carbon film-covered copper grid (Società Italiana Chimici, Rome, Italy) for 2 minutes. To form a thin film specimen the dispersion was blotted from the grid with filter paper, stained with a phosphotungstic acid solution 1% w/v in sterile water. Finally, when the samples were dried after 3 minutes they were examined under the microscope. (S1 Fig)

### Light source

The light source was a diode laser device provided by the RAADMED company from Iran at the wavelength of 450 nm with the output power of 75mW. The radiation spectrum of the diode source (SHARP GH04580A2G) is in the range of 440–460 nm (centered at 450 nm). The distance of sample from the light source was fixed at 6cm with the beam area of light as 6cm$^2$. MEFs were irradiated one time after passing 24 hours from seeding in a 96-well culture plate (the diameter of 7mm) for MTT assay and within a petri dish with a diameter of 35mm for the rest of experiments. The irradiation was performed in a dark room. The irradiation time was automatically adjusted by the device just by setting the energy (Joule) of the radiation due to applying a constant power by the laser (Energy = Power x Time). Therefore, the cells were irradiated for 224 seconds for getting the energy of 17.9 Joule (with energy density of 0.63 J/cm$^2$) and 337 seconds for the dose of 26.9 Joule (energy density of 0.95 J/cm$^2$). For other doses, the time was set in the same way.

### Absorption and UV–Vis spectra

The blank absorption was measured from a pure solvent sample (PBS for nano curcumin and culture medium for MEF cells) using a UV-Vis spectrophotometer (Photonix Ar, Iran). This

helps for removing the light losses due to scattering or absorption by the solvent. Then, the absorbance of various samples was measured (S2 and S3 Figs).

## Cell culture

Mouse embryonic fibroblasts (MEFs) (as a primary culture) was purchased from Royan Institute (Tehran, Iran), which was prepared from a 12.5-day-old mouse from NMRI mouse embryos. MEFs were cultivated in 25-cm$^2$ flasks (TPP, Trasadingen, Switzerland) using the DMEM (Dulbecco's modified Eagle's medium) culture medium (Gibco, USA, Catalog Number: 12-800-082) supplemented with 10% Fetal Bovine Serum (FBS) (Gibco, USA, Product code: 11523387) and penicillin-streptomycin 1% (Gibco, USA, Catalog number: 15140148) maintained at 37˚C in a humidified atmosphere of 95% and 5% CO2 incubator.

## MTT assay

The MTT assay is based on the capacity of mitochondria succinate dehydrogenase enzymes in living cells to reduce the yellow water-soluble substrate MTT (3-[4,5-dimethylthiazol-2-yl]-2,5-diphenyltetrazolium bromide) (5mg.mL$^{-1}$ in PBS) (Sigma-Aldrich, USA) into an insoluble sediment purple formazan. The intensity of the color produced after dissolving the deposition of formazan in organic solvents such as dimethyl sulfoxide is measured spectrophoto-metrically. The reduction of MTT can only occur in metabolically active cells, so this assay represents the level of activity the cells [43]. MEFs were cultured at a density of $4 \times 10^3$ cells per well of the 96-well plates (S4 Fig). After 24 hours, cells were treated with different doses of nanocurcumin (0.25–10 μM) and laser (0.31–25.47 j/cm$^2$) alone and in combination. After 48 hours, MTT solution (20 μl of 5mg. mL$^{-1}$ MTT solution) was added to each well, and plates were incubated at 37˚C for 3–4 hours. Then, the medium was discarded, and formazan crystals were dissolved by adding 200 μl of dimethyl sulfoxide (Sigma-Aldrich, USA). The optical density was measured at 570 nm using a microplate scanning spectrophotometer (BioTek, USA).

## RNA extraction

In each experimental group (DNC alone (0.5, 0.75 μM), LLLT 450 nm alone (0.07, 0.10, 0.63 and 0.95 j/cm$^2$) and the combination group (DNC (0.5, 0.75 μM) + LLLT 450 nm (0.95 j/cm$^2$)), the total RNA was extracted 48h after the final treatment, using RiboEX (GeneAll) according to the manufacturer's instructions. After the extraction, the RNA was quantified in spectrophotometer NanoDrop Lite (Thermo Scientific, USA). The RNA quality was analyzed by agarose gel electrophoresis (BioER, China).

## cDNA synthesis

In order to remove any genomic traces, DNaseI treatment (Takara, Japan) was performed prior to cDNA synthesis at 37˚C for 30 minutes followed by heat and ethylenediamine tetra acetic acid (EDTA, Sigma, USA) to inactivate the DNase enzyme. The synthesis of the complementary DNA (cDNA) was performed using Primescript™RT Reagent Kit (Biofact, Korea) according to the manufacturer's protocol.

## Real-time PCR

The primers for the target genes, TGF-β, VEGF, TNF-α, IL-6 and glyceraldehydes 3-phosphate dehydrogenase (GAPDH) were designed using the Oligo7 software and NCBI database, the sequences of each primer have been shown in S1 Table. Quantitative real-time PCR was performed using standard protocols on a light cycler instrument (Applied Biosystems 7500, USA)

using 2x RT$^2$ SYBR Green High ROX Master mix (Biofact, Korea). The acquired threshold cycle (Ct) values were processed for latter assessments base on the comparative Ct method. The Expression levels of target genes were normalized to the housekeeping gene, GAPDH. The alteration in the expression level of each mRNA was evaluated and normalized in relation to the control group.

## Enzyme-linked immunosorbent assay analysis

Total concentration of proteins under study in MEFs culture supernatant was measured using a sandwich ELISA kits (DuoSet® ELISA Development Systems, U.S.A.). The cell culture supernatant was collected from treated and untreated groups and stored at − 80˚C. The ELISA was performed according to the manufacturer's instructions and analyzed at 450 nm using a 96-well plate reader (BioTek, USA). Standard curve was generated by ODs from serial dilution of a standard protein and corresponding known concentrations. ELISA data of samples from standard curve was used to calculate the concentrations of target proteins in samples. Graph-Pad Prism (version 8) and also Microsoft Excel 2016 were used for data analysis.

## In vitro migration assay

For the purpose of cell motility investigation, MEFs were seeded on Petri dish with a diameter of 35mm. After reaching confluency, the monolayer of cells was subjected to a longitudinal scratch caused by a micropipette tip. Subsequently, medium was carefully replaced with fresh medium to remove debris. MEFs in each group received the respective treatment (optimal doses of nanocurcumin (0.5, 0.75 μM) and low level laser therapy 450 nm (0.95 j/cm$^2$) alone and combinationaly (0.5 μM DNC+ 0.95 j/cm$^2$ LLLT and 0.75 μM DNC+0.95 j/cm$^2$ LLLT)); the control group received no treatment as well. Cells were incubated at 37˚C in incubator (5% CO2). After that, migrated cells into the cell-free area over every 12 hours was evaluated using photographs taken with inverted microscope (Olympus, Japan).

## Cell cycle analysis by flow cytometry

Cell cycle analysis by quantitation of DNA content stained by DNA-binding was performed. Assuming that cells in S phase would have more DNA than cells in G1 and the cells in G2 will be approximately twice as bright as cells in G1 [44]. Accordingly, after 48h treatment of MEFs seeded on Petri dish (diameter of 35mm) with optimal doses of nanocurcumin, laser and combinationaly, the medium culture was removed and cells were harvested by trypsin enzyme, centrifuged and washed with cold PBS. Next, cells were fixed in 1 ml ice-cold 70% ethanol solution (Bidestan, Iran) for at least 30 minutes at 4˚C. each sample, suspended in 500 μl cold hypotonic staining solution containing 50 μg/ml propidium iodide (PI, Sigma, USA), 0.1% Triton X-100 (Merck, Germany), 100 μg/ml RNasea A (Sigma, USA), and 0.1% sodium citrate solution. Number of cells in each cell cycle phase was determined by flow cytometry (BD FACSCanto II, BD Bioscience, San Diego, CA, USA). Data were analyzed using Flowjo software (version 7.6.1).

## Measurement of intracellular ROS

The production and accumulation of intracellular ROS in mouse embryonic fibroblasts (MEFs) cultured on petri dish (diameter of 35mm) and treated in different groups was measured by flow cytometry technique using 2',7'-dichlorodihydrofluorescein diacetate (DCFH-DA) (Sigma-Aldrich, USA). After cell entry, DCFH-DA underwent deacetylation process by intracellular esterases, and subsequently, the presence of intracellular ROS led to the

oxidization of this molecule to form highly fluorescent DCF [45]. Briefly, after 48h, the cells receiving the treatments for each group were trypsinized and centrifugation, the cell pellet was washed with PBS and incubated with 10 μM DCFH-DA for 45 minutes at 37˚C. Then cells were pelleted and washed twice with PBS and subjected to flow cytometry for determination of ROS (BD FACSCanto II, BD Bioscience, San Diego, CA, USA). Data were analyzed using Flowjo software (version 7.6.1).

## Statistical analysis

All the experiments were carried out in triplicate (three times). Statistical analysis were performed using GraphPad Prism software (version 8), and data were displayed as mean ± standard error of means. The statistically significant differences between more than two experimental groups were examined by one-way analysis of variance (ANOVA) was applied to for comparing multiple groups. Consequently, p-values ⏢ 0.05 was considered statistically significant.

## Results

### Effects of DNC and LLLT on MEFs proliferation

The typical MTT assay was employed to obtain the proliferation-inducing doses of DNC and LLLT each alone and in combination on MEFs. Since the aforementioned results were in line with the cell cycle assay results, the outcome was considered as the proliferation-inducing effect. Results revealed that the viability, mitochondrial activity, and growth rate of MEFs treated by increasing dosages of DNC and LLLT (450 nm) each alone was significantly increased sequentially at 0.5 & 0.75μM concentrations of DNC and 0.63 & 0.95 j/cm$^2$ doses of LLLT. Present study also evaluated the proliferative doses of DNC and LLLT in combination on MEFs. Concurrent treatment of DNC (0.5 & 0.75μM) with 0.95 j/cm$^2$ dose of LLLT exerted a notable proliferation in mouse embryonic fibroblasts (Fig 1A–1C).

### Effects of DNC treatment on TGF-β, VEGF, TNF-α and IL-6 expressions in MEFs

To investigate the obtained proliferative concentrations of DNC (0.5, 0.75 μM) on the expression of growth factors (TGF-β, VEGF) and pro-inflammatory cytokines (TNF-α, IL-6) in mouse embryonic fibroblasts, the mRNA expression of these genes were quantitatively evaluated by real-time PCR. The results of gene expression revealed that 48h treatment with dose of 0.75 μM DNC significantly increased the growth factors (TGF-β, VEGF) transcripts in comparison to the untreated group. As expected, a meaningful decrease in the inflammatory factors (TNF-α, IL-6) transcripts was observed after 48h treatment by 0.5 & 0.75 μM DNC. Likewise, both concentrations of 0.5 and 0.75 μM were chosen for further experiments (Fig 2A–2D).

### Effects of LLLT (450 nm) exposure on TGF-β, VEGF, TNF- α, IL-6 gene expressions in MEFs

Inquiring about optimal doses of LLLT (450 nm) on the growth factors and inflammatory cytokines expressions, we employed four doses including 0.07, 0.10, 0.63 and 0.95 j/cm$^2$. We quantitatively evaluated the transcript alterations of these genes by real-time PCR. The results of gene expression displayed that 48h exposure to the doses of 0.63 and 0.95 j/cm$^2$ significantly increased TGF-β expression while doses of 0.10 and 0.95 j/cm$^2$ were able to increase VEGF expression significantly. On the other hand, a significant diminish in the expression of

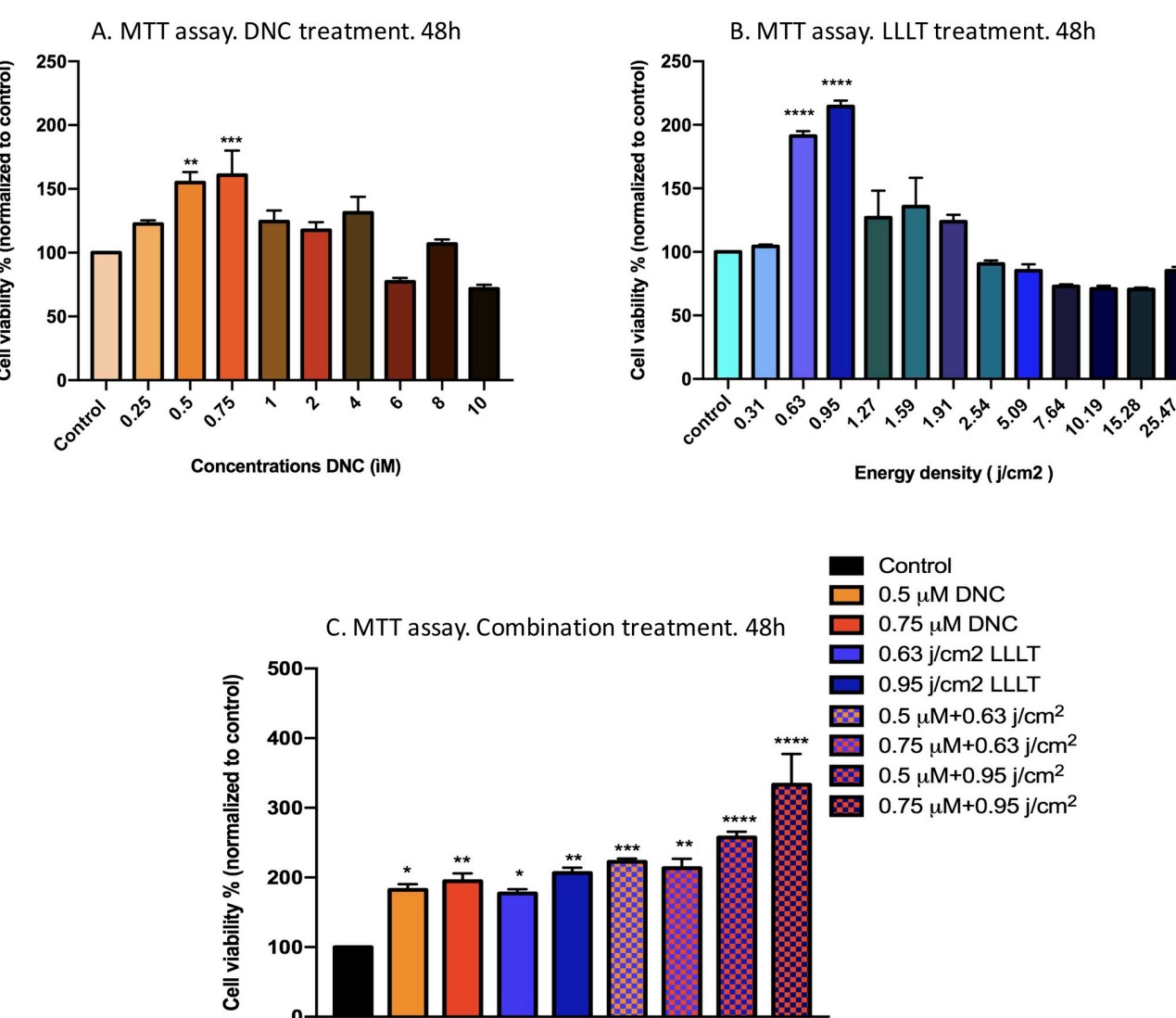

**Fig 1. Comparative analysis of survival and growth rate of MEFs under the treatment of increasing doses of (A) Dendrosomal Nano Curcumin (0.25–10 μM) and (B) low-level laser therapy 450 nm (0.31–25.47 j/cm²) and (C) Combination treatment with optimal doses of DNC and LLLT, after 48h investigated by MTT assay.** [*] Indicates that there is a significant difference between the experimental and control groups. Data are reported as mean ± SEM; n = 3 (*: P < 0.05, **: P< 0.01, ***: P<0.001, ****: P<0.0001).

inflammatory factors (TNF-α, IL-6) 48h post treatment with all four doses of 0.07, 0.10, 0.63 and 0.95 j/cm² was observed. Therefore, dose of 0.95 j/cm² of LLLT (450 nm), which nearly exerted the expected effect to genes under the study were selected for subsequent experiments (Fig 3A–3D).

## Optimal doses of DNC and LLLT in combination alters the expression of TGF-β, VEGF, TNF-α and IL-6 in MEFs

To investigate the difference in expression of growth factors (TGF-β, VEGF) and inflammatory cytokines (TNF-α, IL-6) alone and co-treated with DNC+LLLT simultaneously and with intervals, the mRNA expression of these genes were quantitatively evaluated by real-time PCR. The results indicate a significant increase in growth factors (TGF-β, VEGF) and a significant

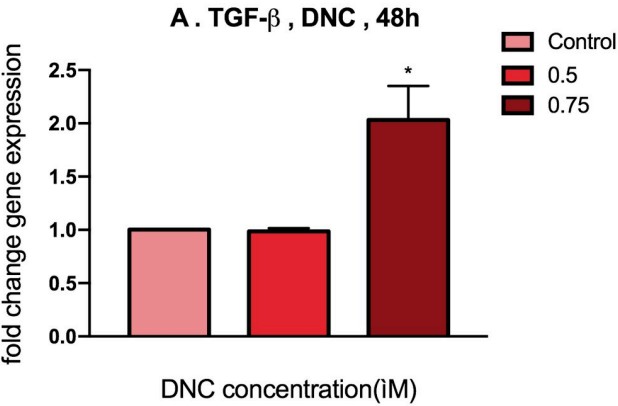

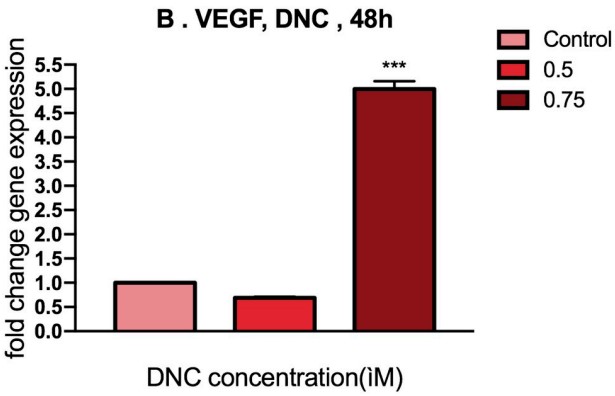

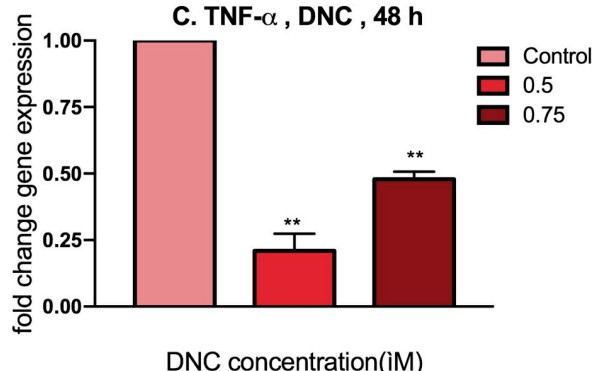

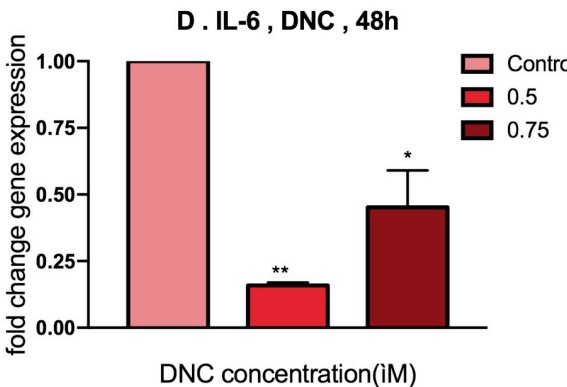

**Fig 2. TGF-β, VEGF, TNF-α, IL-6 genes expression in MEFs under 48h treatment with doses of 0.5, 0.75 μM DNC.** Charts Represents a significant increase in TGF-β (A) and VEGF (B) and a significant decrease in TNF-α (C) and IL-6 (D) gene expression in MEFs compared with control group. Data are shown as mean ± SEM; n = 3 (*: $p < 0.05$, **p: $< 0.01$, ***p: $< 0.001$).

decline in pro-inflammatory cytokines (TNF-α, IL-6) of the MEFs under simultaneous treatment of DNC+LLLT (0.5 μM +0.95 j/cm$^2$) synergistically in comparison to the DNC and LLLT treated groups each alone and the control group. It should be noted that concurrent combination therapy, exerted more favorable and more tangible expression changes in the genes under the study compared to the group exposed to the 0.95 j/cm$^2$ dose of LLLT (450 nm) 4 hours later than DNC treatment. Therefore, synchronous combination therapy was selected for subsequent experiments (Fig 4A–4D).

## Simultaneous treatment with optimal doses of DNC and LLLT altered the secreted level of TGF-β and IL-6 proteins from MEFs

To assess whether optimal doses of DNC (0.5, 0.75 μM) and LLLT 450 nm (0.95 j/cm$^2$) alone and in combination can enrich the secreted level of growth factors as well as downgrade the secretion of inflammatory elements involve in wound healing, the protein concentration of TGF-β as a growth factor and IL-6 a pro-inflammatory cytokine were detected in MEFs culture supernatant by sandwich ELISA. The results indicate a significant degrade in IL-6 production

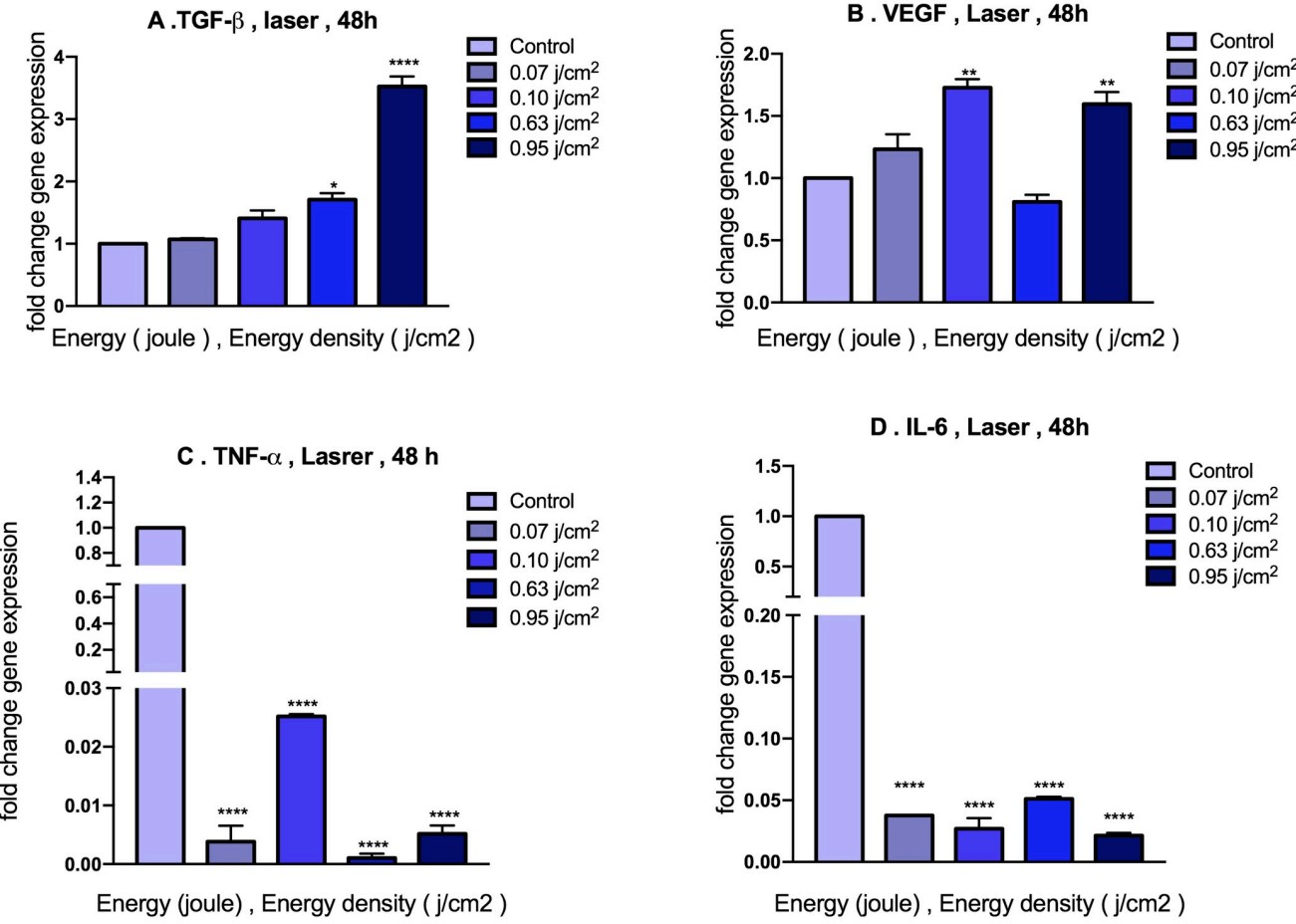

**Fig 3. Alteration expression of TGF-β, VEGF, TNF-α, IL-6 in MEFs under 48h treatment with doses of 0.07, 0.10, 0.63, 0.95 j/cm² LLLT (450 nm).** (A&B) represent a significant increase in growth factors expressions and (C&D) represent a significant multifold decrease in pro-inflammatory cytokine transcripts after 48h exposure to LLLT compared with control group. Data are represented as mean ± SEM; n = 3 (*: $p < 0.05$, **: $p < 0.01$, ****: $p < 0.0001$).

and secretion in MEFs treated with optimal doses of DNC (0.5, 0.75 μM) and LLLT 450 nm (0.95 j/cm²) alone and concurrent combination therapy (0.5 μM DNC + 0.95 j/cm² LLLT) compared to the control group. Nevertheless, no statistically significant difference was observed in IL-6 production in MEFs treated with combination therapy (0.5 μM DNC + 0.95 j/cm² LLLT) compared to DNC (0.5, 0.75 μM) and LLLT 450 nm (0.95 j/cm²) alone. Contrary to the IL-6, the combinational therapy (0.75 μM DNC+ 0.95 j/cm² LLLT at the same time) brought about a significant increase in TGF-β production and secretion in MEFs compared to the other treated and control group and results confirmed our gene expression data (Fig 5A and 5B).

## The effects of DNC, LLLT and combination treatment on MEFs migration

Scratch-wound assay as an in vitro migration model was employed to analyze the effects of DNC, LLLT and their combination on the migration of mouse embryonic fibroblasts. For excluding any possible effect of proliferation on migration, the effects of different treatments tested were assessed on migration of MEFs prior to their doubling time (24 h), and images were taken every 12 hours after scratch gap creation on cell monolayers. The results pointed

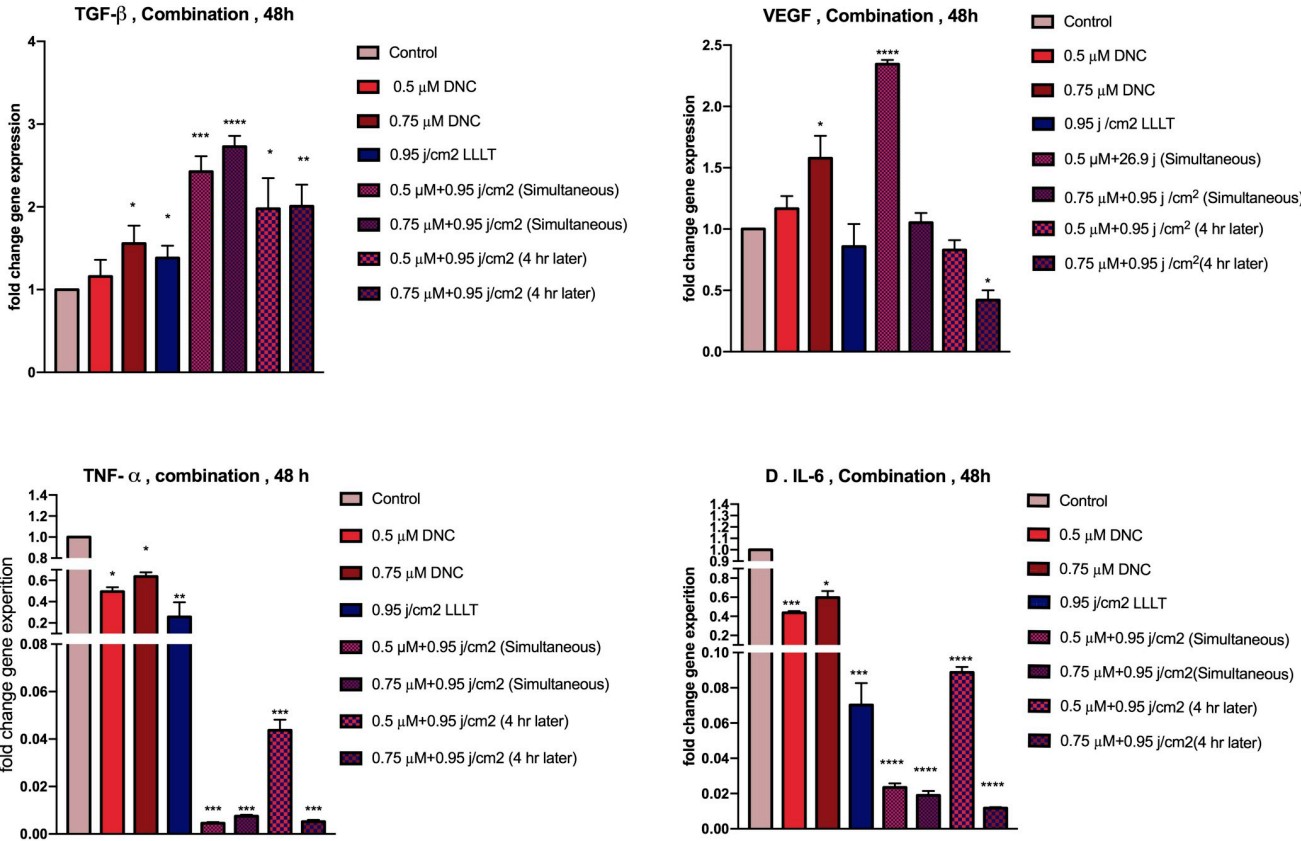

**Fig 4. Gene expression in MEFs under 48h treatment with DNC (0.5, 0.75 μM) and LLLT 450 nm (0.95 j/cm²) alone and in combination.** (A&B) Represent a significant increase in TGF-β and VEGF transcripts and (C&D) represent a significant decline in TNF-α and IL-6 expression after treatment with DNC+LLLT simultaneously and with 4h time intervals similarly. Data are presented as mean ± SEM; n = 3 (*: P < 0.05, **: P< 0.01, ***: P<0.001, ****: P<0.0001).

out that migration of cells to the bared area on both sides of the scratch in the groups 12 and 24h post treated with optimal doses of DNC (0.5, 0.75 μM) and LLLT 450 nm (0.95 j/cm²) alone was more than the control group. Furthermore, MEFs treated with concurrent combination therapy (0.5 μM DNC+ 0.95 j/cm² LLLT and 0.75 μM DNC+0.95 j/cm² LLLT) exhibited

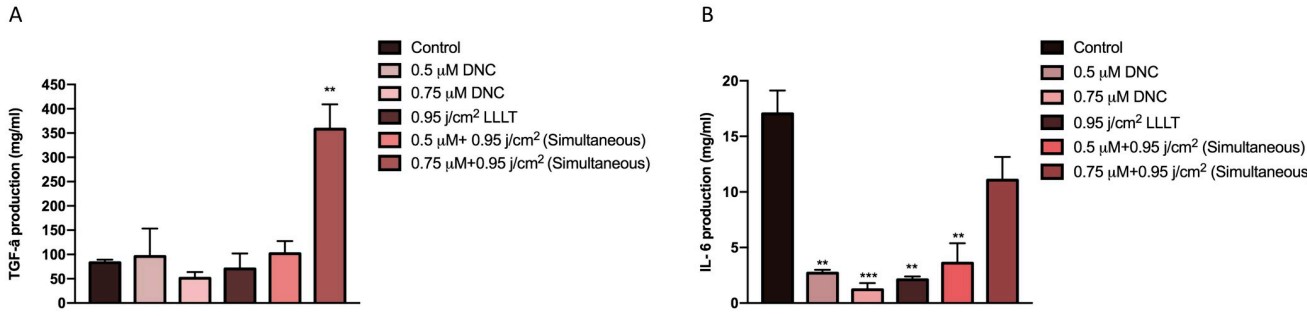

**Fig 5. Quantification of TGF-β and IL-6 protein levels in cell culture supernatant from MEFs after 48h treatment with DNC and LLLT 450 nm and combination treatment using sandwich ELISA.** (A) TGF-β, was significantly increased in the MEF treated with combination treatment (0.75 μM DNC + 0.95 j/cm² LLLT at the same time) compared to the control and other groups. (B) IL-6, was significantly decreased in the MEF treated with DNC and LLLT alone and in combination compared to control group. Data are demonstrated as mean ± SEM; n = 3 (**: P< 0.01, ***: P<0.001).

more dramatic increase in migration toward denuded area than cells treated with optimal doses of DNC (0.5, 0.75 μM) and LLLT 450 nm (0.95 j/cm$^2$) separately. Also, the migration continued time dependently from 12 to 24h (Fig 6A and 6B).

## DNC and LLLT treatment alone and in combination affect MEFs cell cycle through increasing the S phase

The ability of optimized doses of DNC, LLLT alone and in combination to modulate the cell cycle was investigated. In the presence of 0.5, 0.75 μM of DNC, 0.95 j/cm$^2$ of LLLT (450 nm) and combination treatment (0.5 μM DNC+0.95 j/cm$^2$ LLLT and 0.75 μM DNC+ 0.95 j/cm$^2$ LLLT at the same time) a significant increase in the population of cells in the S phase and a meaningful decrease in the sub-G0/G1 population of MEFs compared to the control group was observed. It should be noted that, there was no significant difference between different treated groups in S phase enhancement and sub-G0/G1 phase decrement (Fig 7A–7C).

## DNC+ LLLT treatment can regulate the intracellular ROS accumulation in MEFs

Since extended oxidative stress generally inhibits proper wound healing, controlling the produced ROS may improve the aforementioned process. In spite of a significant increase in the mean DCF fluorescence intensity in MEFs treated with 0.95 j/cm$^2$ of LLLT (450 nm) after 48h compared to the control and other treated groups, there was no significant alteration in the groups treated with DNC (0.5, 0.75 μM) and combination therapy (0.5 μM DNC + 0.95 j/cm$^2$ LLLT and 0.75 μM DNC + 0.95 j/cm$^2$ LLLT at the same time) compared to the control group. These results indicate that anti-oxidant doses of DNC could suppress the laser-induced excessive enhancement in intracellular ROS levels and oxidative stress in MEF cells (Fig 8A–8C).

## Discussion

Pressure ulcers are one of most dominant problems occurring in patients with mobility limitations. Affecting numerous people around the world and imposing considerable medical expenses on both individuals and health services of countries each year [46, 47], this challenge has attracted considerations. Although several therapeutic approaches are using in order to heal such wounds [8, 9, 48], there is still obstacles in appropriate healing which necessitates new practical alternative or adjunctive. Plenty of studies done by researches have shown the potent beneficial effects of curcumin as a herbal medicine as well as low-level laser therapy (LLLT) as novel therapeutic approaches [20, 35, 49]. Since curcumin works on different stages of the wound healing process [17, 18], we sought to investigate the anti-oxidant, anti-inflammation and proliferative effects of curcumin along with LLLT on important events in wound healing process including proliferation, migration and alteration of cytokines and growth factors involved as well as modulating the oxidative stress, through in vitro studies on mouse embryonic fibroblast cells (Table 1). In order to overcome curcumin's poor bioavailability and to get the maximum therapeutical effects of it, we utilized dendrosome nanoparticles in the present study.

Previous studies support the stimulatory effects of curcumin at low concentrations [18, 19, 50] and LLLT at different wavelengths (red 620–770 nm, blue 400–480 nm, green 470–550 nm) with very low doses on cell growth and proliferation rate of fibroblasts [51, 52]. In line with our hypothesis, the obtained results from MTT along with cell cycle assays demonstrated that dendrosomal nano curcumin (DNC) at antioxidant concentrations (0.5, 0.75μM) as well as low doses of LLLT 450 nm (0.63, 0.95 j/cm$^2$) were able to promote growth and proliferation

**A**

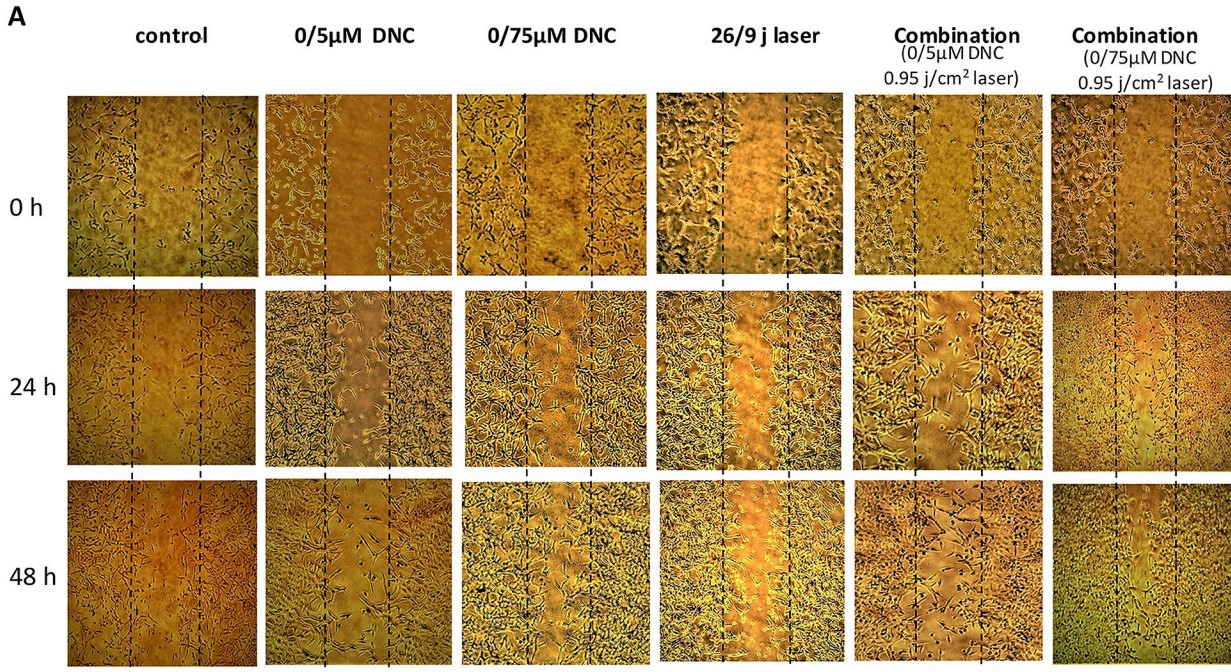

**B**

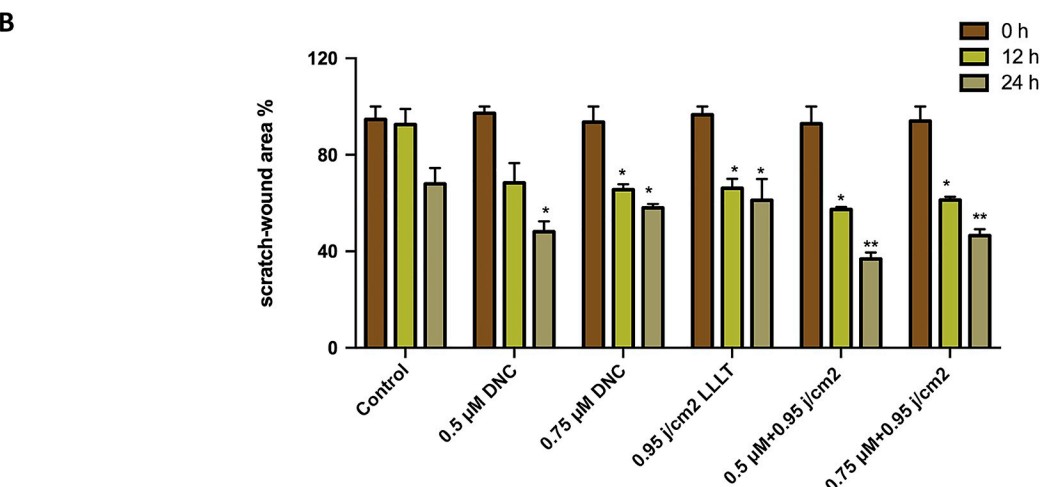

**Fig 6. Migratory ability of MEFs 12 & 24h post treated with DNC and LLLT 450 nm and combination treatment.** (A) Representative images (40× magnification) from wound healing assay of MEFs treated with optimal doses of DNC, LLLT 450 nm and in combination at different time points (0, 12 and 24 h). Control groups were not treated. Cell migration into the cell-free region was marked by dotted lines at 0 hours (scale bar: 750μm). (B) Comparing the migrated percentage of MEFs with the cell-covered area at indicated time points using ImageJ software illustrates the wound healing rate represented in bar graph. Data are expressed as the mean± SEM of triplicate experiments (*: $P < 0.05$; **: $P < 0.01$).

in MEFs; however, combination treatment exerted more obvious proliferative effect on mouse embryonic fibroblasts. Therefore, it can be said that increasing the rate of growth and proliferation of fibroblasts in the wound microenvironment, may contribute to fill the denuded area and replace the lost cells at the site of the lesion as well as releasing cytokines and growth factors required for proper healing [53].

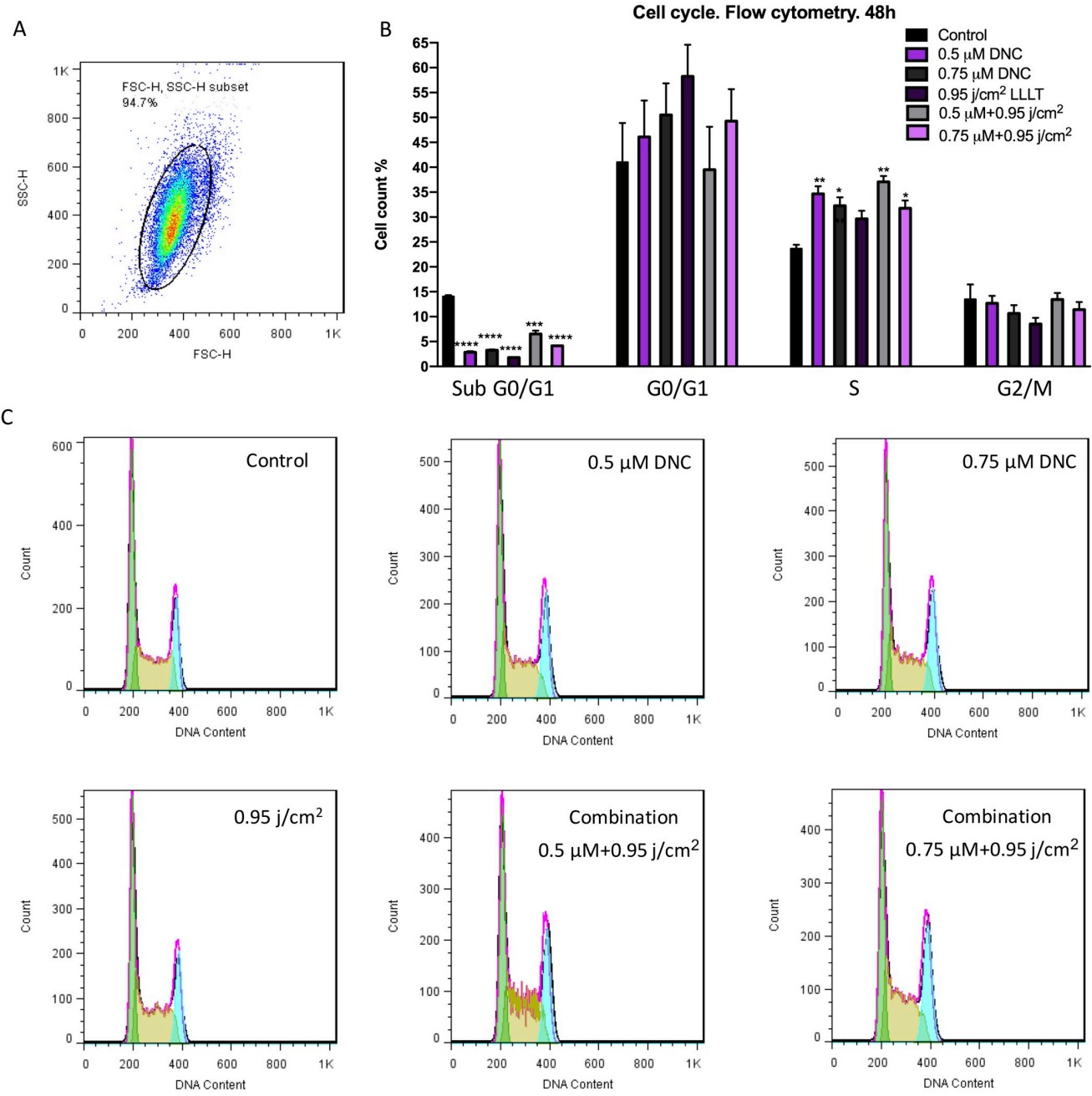

**Fig 7. Flow cytometry analysis of cell cycle in MEFs after 48h post-treated with optimal doses of DNC, LLLT (450 nm) and combination treatment.** (A) Graph represents the cell population under the study using FlowJo Software. (B) Cell cycle distribution of DNC, LLLT and DNC+LLLT treated MEFs after 48h. (C) Histogram of cell cycle analysis of MEFs exposed with DNC, LLLT and combination treatment after 48h. The results are shown as the mean± SEM of triplicate experiments (*: P < 0.05, **: P< 0.01, ***: P<0.001, ****: P<0.0001).

Cell proliferation is a fundamental process by which the cell cycle is executed from G1 phase, entering the subsequent S phase and G2/M phase thereafter [54]. In 2015, a study reported an increase in the population of keratinocyte cells at the G2/M transition point, which represents an increase in cell proliferation when treated with low concentrations of curcumin along with red and blue LED light [13]. To verify the influence of DNC (0.5, 0.75 μM)

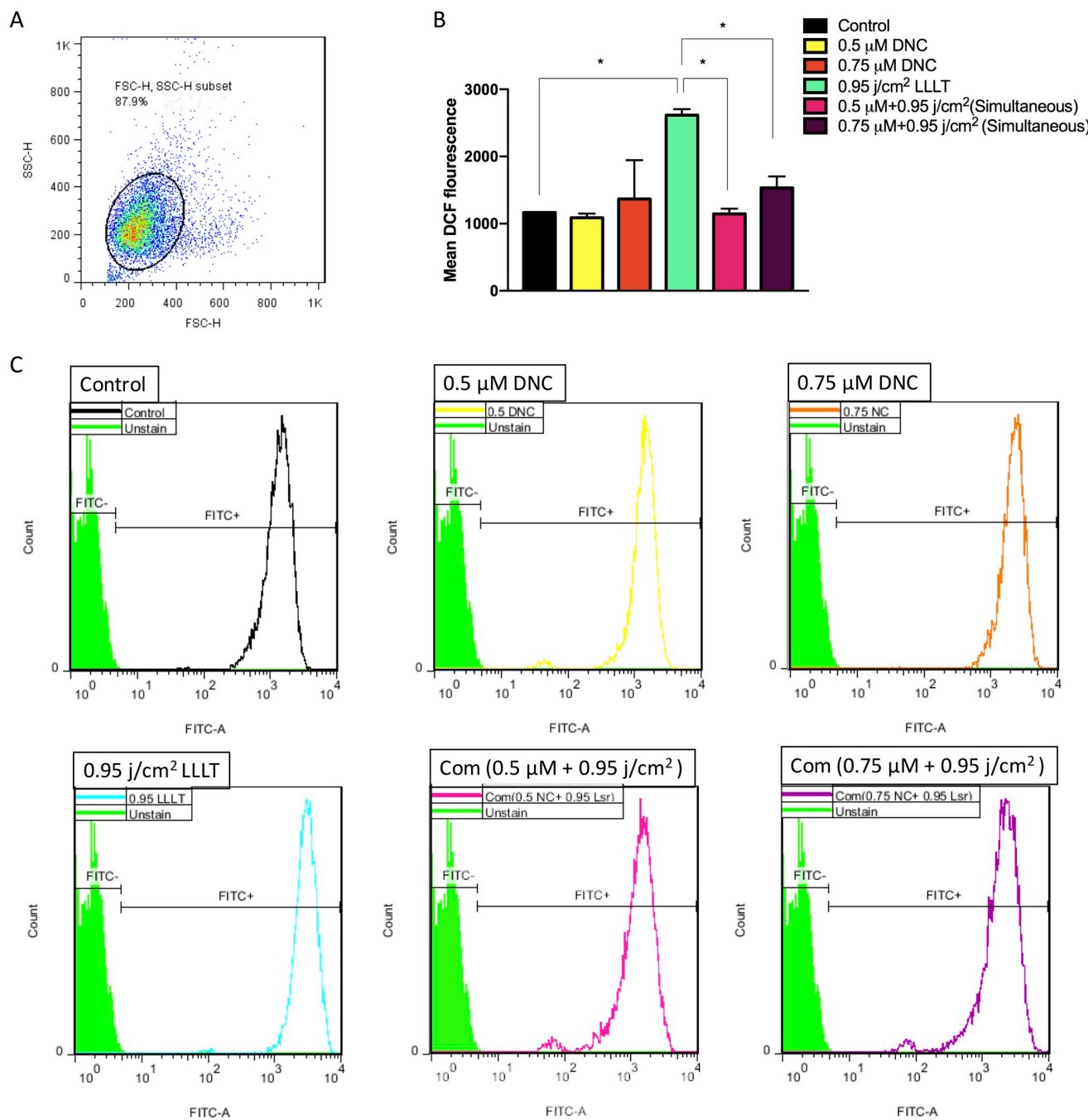

**Fig 8. Flow cytometry analysis of intracellular ROS generation in MEFs under 48h treatment with DNC, LLLT (450 nm) and in combination.** (A) Graph represents the cell population under the study using FITC-A versus SSC-A plot. (B) Bar Graphs showing the difference in DCF fluorescent intensity in MEFs, each bar demonstrates the mean ± SEM, n = 3 ($^*$p < 0.05). (C) Histograms represent changes in the intensity of DCF fluorescence, arising from production and accumulation of intracellular ROS in MEFs.

and LLLT 450 nm (0.95 j/cm$^2$) alone and in combination on cell cycle regulation of MEFs during wound healing, DNA content of cells were evaluated in three major cell phases (G1, S, G2/ M). In accordance with the results obtained from MTT assay, the cell population under treatment with DNC and LLLT 450 nm each alone and concurrently was enriched in the S phase

**Table 1. The comparison of our findings with the previous research results in more detail.**

| | Concentration / Dose | Effect | Reference |
|---|---|---|---|
| Curcumin | 0.06–1 | Proliferation/Migration | [45, 16] |
| LLLT | Red 620–770 nm/ 0.3 J/cm$^2$, Blue 400–480 nm/ 0.3 J/cm$^2$, Green 470–550 nm/ 0.3 J/cm$^2$ | Proliferation/ Migration | [46, 47] |
| DNC+LLLT | 0.5, 0.75µM + Blue 450 nm/ 0.63, 0.95 j/cm$^2$ | Proliferation/Migration | |
| Curcumin | 0.3% on day 3 | Increase in TGF-β | [52] |
| LLLT | 2 J/cm$^2$ | Increase in TGF-β | [31] |
| DNC+LLLT | 0.75µM + Blue 450 nm 0.95 j/cm$^2$ | Increase in TGF-β | |
| Curcumin | | Increase in VEGF | [36, 55] |
| LLLT | Red 620–770 nm/ 0.3 J/cm$^2$, Blue 400–480 nm/ 0.3 J/cm$^2$, Green 470-550nm/ 0.3 J/cm$^2$ | Increase in VEGF | [46] |
| DNC+LLLT | 0.75µM + Blue 450 nm 0.95 j/cm$^2$ | Increase in VEGF | |
| Curcumin | 700 µg/well for 1–3 days | Decrease in TNF-α & IL-6 | [59, 61] |
| LLLT | Blue 450 nm 25, 50, 100 mJ/cm$^2$ | Decrease in TNF-α & IL-6 | [60] |
| DNC+LLLT | 0.5, 0.75µM + Blue 450 nm 0.95 j/cm$^2$ | Decrease in TNF-α & IL-6 | |
| Curcumin | Anti-oxidant doses | ROS scavenging | [16, 65] |
| LLLT | Blue 456 nm 5 W/m$^2$ | ROS generation | [60] |
| DNC+LLLT | 0.5, 0.75 µM + Blue 450 nm 0.95 j/cm$^2$ | No change in ROS | |

representing progression of cells towards growth and proliferation. Likewise, a significant diminish in the Sub-G0/G1 phase was observed.

One of the most important events contributing to wound healing is the migration of fibroblasts to the site of the wound mediated by chemotaxis agents required for further proliferation and differentiation into myofibroblasts [31, 55]. Using the scratch-wound assay, we showed that cell migration into the cell-free area was evoked by culturing with DNC (0.5, 0.75 µM) and LLLT 450 nm (0.95 j/cm$^2$) exposure. This aspect of our results was consistent with previous reports revealed the increasing effect of curcumin [18, 19, 50] and low-level lasers [37, 51] on cell migration, through activating cellular signaling pathway involved in this process. However, the migration and progression of MEFs towards each other in the denuded area was counted more intense in simultaneous DNC+LLLT 450 nm treatment group so that the created gap was partially filled after 24h.

Among the large number of growth factors required for wound healing, transforming growth factor-β (TGF-β), produced by immune and non-immune cells such as fibroblasts, has the most extensive spectrum of effects including reducing inflammation, stimulating proliferation and migration, initiating angiogenesis, conversion of fibroblasts to myofibroblasts and the production of structural proteins and wound strength [56]. Previous studies have also designated that low doses of curcumin [57, 58] and LLLT (wavelengths ranging from 660 nm to 1064 nm) [34, 59] increased the expression of TGF-β acting as both growth factor and anti-inflammatory cytokine. This study also points out that although the expression of TGF-β was significantly increased in MEFs treated with anti-oxidant dose of dendrosome-coated curcumin (0.75µM) and low doses of LLLT 450 nm (0.95 j/cm$^2$), this enhancement was more intense and noticeable in MEFs receiving the concurrent combination treatment suggesting a synergistic effect in TGF-β transcripts quantity as well as its secretion. It should be noted that, increased expression of this growth factor was greater in cells treated with the combination therapy simultaneously (DNC and LLLT treatment at the same time) than combination therapy of 4 hours apart (laser irradiation 4 hours after DNC treatment).

Formerly, researchers have shown that curcumin [39, 60] and laser therapy [43, 51] at specific doses and under certain conditions were able to enhance angiogenesis by increasing the expression of the most important pro-angiogenic ligand called VEGF specifically in the

influential cells (fibroblasts) producing this ligand for endothelial cells at the site of the lesion. VEGF has been well known for its pro-angiogenic potency through stimulation of endothelial cell proliferation, migration, and tube formation activities [45]. According to the results obtained, a significant increase in VEGF expression post-treated with DNC (0.75μM) and LLLT 450 nm (0.95 j/cm$^2$) was perceived. In addition, the observations of this study suggest that the increased expression of this gene in simultaneous combination treatment was more pronounced.

Inflammation has been considered as the vital second phase of the wound healing process manipulated by cytokines and growth factors. While key inflammatory cells at the site of injury assumed to be the immune cells that clean the wound from infection and damaged structural proteins as well as play chemotactic role in inviting fibroblasts and other cells involved, damaged fibroblasts from nonhealing wounds have been shown to produce higher levels of pro-inflammatory cytokines and represent poor responses to growth factors as well. Previous reports have shown that prolonged up-regulation of inflammatory cytokines such as TNF-α which induces IL-6 production as well can cause local inflammation and persistent localized inflammation have the potency to damage the tissue [31]. Inevitably, controlling the prolonged inflammation may optimize the wound healing process accelerating the proliferation phase entrance [61–63]. Many researchers have demonstrated that curcumin, as well as blue light at 450 nm at very low doses, modulates inflammation by reducing the expression of pro-inflammatory cytokines including TNF-α and IL-6 [18, 64, 65]. Herein, we obtained a significant decline in the expression of TNF-α and IL-6 in MEFs under treatment of DNC (0.5, 0.75 μM) and LLLT 450 nm (0.95 j/cm$^2$). Remarkably, a more pronounced decrease in expression of these two factors was observed in simultaneous combination treatment compared to DNC and LLLT groups alone and the control group. According to previous research studies, blue 470 nm light irradiation [37] and curcumin [66] revealed a decrease in IL-6 concentrations compared to the non-treated groups. In line with these observations, lower concentrations of IL-6 secreted from MEFs were found following the uptake of DNC and LLLT 450 nm exposure both alone and in combination.

The production of ROS such as superoxide radical ($O_2^-$), hydroxyl radical (OH) and hydrogen peroxide ($H_2O_2$) by aerobic metabolism at a certain level is natural and [67] executing intracellular secondary messenger roles in signaling pathways specially the ones involved in wound healing as well as being required for immune system defense against pathogens. However, excessive accumulation of intracellular ROS is considered to be the major cause of inflammation during wound healing activity; Cause damage to cellular components, delay the wound healing process and cause non-healing ulcers [38, 39]. Previous studies have reported that laser irradiation within 456–808 nm spectrum had generated a high concentration of ROS in various cells primarily induced signaling pathways required for proper wound healing however, the prolonged production and accumulation of that resulted in oxidative stress [68, 69]. On the other hand, it had been shown that curcumin with 10μM anti-oxidant concentration had the potency to scavenge the intracellular ROS [18, 69]. Herein, we interrogated the anti-oxidative efficacy of DNC against LLLT-induced ROS accumulation. By reducing the oxidative stress caused by LLLT, DNC displayed its potency to modulate proper wound healing since the excessive accumulated ROS has always been considered as the main reason of cell damage and inflammation during wound healing activity. In other words, the combination therapy (DNC+LLLT) could be a regulator in production and accumulation of ROS for proper wound healing.

## Conclusion

The molecular and cellular events in fibroblasts as one of the most important cells involved in wound healing process, were significantly modulated by concurrent exposure to anti-oxidant

concentrations of dendrosome-encapsulated curcumin and low-level laser therapy 450 nm which improve the wound healing activity. The results pointed out that this combination treatment enriched S phase entry and therefor increased fibroblasts proliferation as well as migration and shortening the inflammatory phase. These events could be possible through up-regulation of growth factors and anti-inflammatory cytokines (such as VEGF and TGF-β) and modulation of inflammatory cytokines (especially IL-6 and TNF-α). It should be noted that in combination treatment, nanocurcumin were able to prevent excessive production and accumulation of the laser-induced ROS. Although further in vivo studies are required, it is worth-noting that this combination treatment has the potency to treat pressure ulcers.

## Supporting information

**S1 File.**
(DOCX)

**S1 Fig. DNC characterization.** (A): DLS diagram of dendrosomal nanocurcumin (DNC) in which was analyzed in terms of size distribution by intensity of light scattering. With regard to DLS measurements (hydrodynamic radius), dendrosome particles size are majorly around 100 nm. (B): The size, ζ-potential and polydispersity index (PDI) of DNC. (C): Transmission electron micrographs (TEM) of DNC: spherical shape of DNC which indicate the compact form of micelles-carriers.
(JPG)

**S2 Fig. The UV–Vis spectra of DNC.** The UV–Vis spectra of Dendrosomal Nano Curcumin (DNC) dissolved in PBS with concentrations of 5 and 10 μM (anti-oxidant doses) represents that its light absorption spectrum is a broad band (300–500 nm) with maximum absorbance peak at a wavelength ~425 nm. Also, the more the DNC solution is diluted, the more the UV–VIS absorption intensity decreases. However, the wavelength of light absorption remains constant.
(JPG)

**S3 Fig. Absorption and UV–Vis spectra of MEFs.** Absorbance measurement by UV–VIS for MEFs suspended in cell culture medium (DMEM) shows that absorption spectrum of MEFs is uniformly a broad band above 300 nm.
(JPG)

**S4 Fig. MTT standard curve.** To optimize the number of MEFs for seeding in 96-well plate, MTT standard curve was performed. MEFs were seeded in 96-well plates at 1000 number intervals from 2000 to 12,000 (4 replicates). After 72 hours, MTT assay test performed without receiving any treatment. The results indicated a linear relationship between the number of cells and the intensity of optical density (OD). Since the aim of this study was to investigate the growth and proliferation of cells under appropriate treatments, half the maximum optical density suitable for MTT assay (OD = 1), can be used for this experiment. Therefore, out of 11 cell groups tested, a cell population of 4000 cells with average absorbance of 0.4–0.6 was selected for further studies.
(JPG)

**S1 Table. Primer sequences used to analyze the gene expression.**
(PDF)

**S1 Raw data. Cell cycle analysis by flow cytometry.**
(JPG)

**S2 Raw data. Measurement of intracellular ROS by flow cytometry.**
(JPG)

**S3 Raw data. MTT assay.**
(PDF)

## Acknowledgments

We thank Dr. Sara Soudi, Department of Immunology, School of Medical Sciences, Tarbiat Modares University, for providing technical support Enzyme-linked immunosorbent assay. The authors like to thank Mohammadjavad Karimi Taheri, Department of Molecular Genetics, Faculty of Biological Sciences, Tarbiat Modares University, for his advice and support in laboratory techniques and experiments. The authors also like to thank Hossein Siampour and Dr. Sara Abbasian from department of Physics, Tarbiat Modares University for set-up the laser device for cells and their advice on the physical aspects of the project. We thank Dr. Ali Dinari, Polymer Reaction Engineering Department, Faculty of Chemical Engineering, Tarbiat Modares University, for his advice and support in Dynamic light scattering (DLS) and Transmission electron micrographs (TEM) of DNC.

## Author Contributions

**Conceptualization:** Afsaneh Ebrahiminaseri, Majid Sadeghizadeh.

**Data curation:** Afsaneh Ebrahiminaseri.

**Formal analysis:** Afsaneh Ebrahiminaseri.

**Investigation:** Afsaneh Ebrahiminaseri.

**Methodology:** Afsaneh Ebrahiminaseri, Ahmad Moshaii, Zohreh Safari.

**Project administration:** Afsaneh Ebrahiminaseri, Majid Sadeghizadeh.

**Resources:** Afsaneh Ebrahiminaseri, Majid Sadeghizadeh.

**Software:** Afsaneh Ebrahiminaseri, Zohreh Safari.

**Supervision:** Majid Sadeghizadeh.

**Validation:** Majid Sadeghizadeh, Ahmad Moshaii.

**Visualization:** Majid Sadeghizadeh.

**Writing – original draft:** Afsaneh Ebrahiminaseri.

**Writing – review & editing:** Afsaneh Ebrahiminaseri, Majid Sadeghizadeh, Ahmad Moshaii, Golareh Asgaritarghi, Zohreh Safari.

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
