## [Decision Letter · Decision Letter 0]

24 Nov 2020

PONE-D-20-33956

Combination treatment of dendrosomal nanocurcumin and low-level laser therapy develops proliferation and migration of mouse embryonic fibroblasts and alter TGF-β, VEGF , TNF-α and IL-6 expressions involved in wound healing process

PLOS ONE

Dear Dr. sadeghizadeh,

Thank you for submitting your manuscript to PLOS ONE. After careful consideration, we feel that it has merit but does not fully meet PLOS ONE’s publication criteria as it currently stands. Therefore, we invite you to submit a revised version of the manuscript that addresses the points raised during the review process.

The reviewers have made several points that need to be addressed. You must have your manuscript professionally edited for scientific English.

We look forward to receiving your revised manuscript.

Kind regards,

Michael R Hamblin

Academic Editor

PLOS ONE

Journal Requirements:

2.Thank you for stating the following in the Acknowledgments Section of your manuscript:

[We would also like to

acknowledge the Iran National Science Foundation (INSF) for its financial support of the

project.]

 [The author(s) received no specific funding for this work.]

Reviewers' comments:

Reviewer's Responses to Questions

**Comments to the Author**

1. Is the manuscript technically sound, and do the data support the conclusions?

Reviewer #1: Partly

Reviewer #2: Yes

2. Has the statistical analysis been performed appropriately and rigorously? 

Reviewer #1: Yes

Reviewer #2: Yes

3. Have the authors made all data underlying the findings in their manuscript fully available?

Reviewer #1: Yes

Reviewer #2: Yes

4. Is the manuscript presented in an intelligible fashion and written in standard English?

Reviewer #1: Yes

Reviewer #2: Yes

5. Review Comments to the Author

Reviewer #1: The concept of this work is interesting, however, the manuscript suffers from significant grammatical and syntax errors, and i strongly suggest editing whole of the manuscript by a native-English speaker before proceed for the publication. Anyway, the major revision needed and the below points should be addressed before any publication:

1. The introduction needs major revision; please add more up-to-date prestigious papers related to the background of your work

2. Regarding the UV absorbance (Fig S1); i guess your sample tube or quartz cell is not clean, or there is problem with your DNC sample. Please re-investigate the UV experiment, and please add x- and y-axis labels to the graphs. Also, i see negative phase absorption in your UV, please have a complete and in-depth discussion regarding this.

3. Regarding the PL (Fig S2); why you state in the x-axis the time? this is the wavelength!!! what is the role of time here? also what is the intensity unit? a.u.?

4. The same question and concerns regarding the UV of Fig s3. arises, please be careful and address them.

5. Regarding the Fig. S4; why you are not trying to remove the lines inside the figure? is odd for me; please clear the graphs and set a clean white background for all of them

6. Regarding Fig. S5; please mark the states with significant changes, i cannot follow your text with the figure changes

7. Why there is no TEM, DLS and FESEM of the used DNC? how can i be sure that your used DNC is in the nano-dimension. You should provide all of the TEM, FESEM and DLS results to prove the nano-sized curcumin.

8. The quality of all of the images are really poor; please replace all of the images with high-quality one.

9. Please compare your results with the literature; it would be better to add a table regarding this

10. Also, i suggest to provide the raw data including the flowcytometry raw datas as well as the raw MTT results in the supplementary information; i need to be sure about the validity of your results

11. Please be careful about the syntax errors: 450nm is wrong!! 450 nm is in the appropriate form.

Major revision

Reviewer #2: This is an interesting study that looks at the proliferation, motility of embryonic fibroblasts and the expression profiles TGF-β,VEGF , TNF-α and IL-6 following application of nanocurcumin and 450 nm light.

Before publication, please consider the following comments.

P 4 line 5: what culture medium was used?

P4 4. Line 6: Although citations were included, some brief details of this optimized protocol should be included.

P4 Line 8: please add the exact model used and/or emission spectrum

P4 line 34: please defined DMEM when first used.

P5. Line 7: paint? Do you mean the pigment?

P5 Line 10: indeed, the MTT assay is a measure of metabolic activity. Metabolism is highly dynamic so I am unsure one could claim that it is strictly associated with viability?

P5 Line 12: what were the doses of nanocurcumin and/or light?

P5. Line 20: which experimental groups? please be specific.

P6. Line 4: were the proteins not extracted? Or was it just the supernatant? I would think it would be difficult to get enough protein concentration this way.

P6 Line 18: the authors need to provide the exact concentrations/doses used.

P 7: Line 13. Were the experiments performed on 3 different days?

6. PLOS authors have the option to publish the peer review history of their article (what does this mean?). If published, this will include your full peer review and any attached files.

Reviewer #1: No

Reviewer #2: No

---

## [Author Response · Author response to Decision Letter 0]

23 Jan 2021

Dear reviewer1

First, we would like to thank you for reviewing our manuscript. We also thank the reviewers for their careful reading and constructive suggestions. Reviewed manuscript was studied and revised according to reviewers’ comments. The comments were responded point by point and highlighted through the body of manuscript (in this file). It is our belief that the manuscript was substantially improved after applying the valuable reviewers’ comments.

Reviewer #1: The concept of this work is interesting; however, the manuscript suffers from significant grammatical and syntax errors, and I strongly suggest editing whole of the manuscript by a native-English speaker before proceed for the publication. Anyway, the major revision needed and the below points should be addressed before any publication:

Author:

Thank you very much for your positive feedback to our manuscript. According to your comment, text was corrected and highlighted.

1. The introduction needs major revision; please add more up-to-date prestigious papers related to the background of your work

Author:

The comment considered, I added papers related to text corrected and highlighted.

2. Regarding the UV absorbance (Fig S1); I guess your sample tube or quartz cell is not clean, or there is problem with your DNC sample. Please re-investigate the UV experiment, and please add x- and y-axis labels to the graphs. Also, I see negative phase absorption in your UV, please have a complete and in-depth discussion regarding this.

Author:

We repeated the UV experiment according to the points you mentioned and the result was a graph without negative phase absorption. Also, I added x- and y-axis labels to the graph.

3. Regarding the PL (Fig S2); why you state in the x-axis the time? this is the wavelength!!! what is the role of time here? also what is the intensity unit? a.u.?

Author:

Thank you for your precise comment. After reading your opinion and reviewing the article again, we came to the conclusion that this section was not necessary and the results of this experiment did not play a role in the overall conclusion of our article, so we deleted it.

4. The same question and concerns regarding the UV of Fig s3. arises, please be careful and address them.

Author:

We repeated the UV experiment according to the points you mentioned and the result was a graph without negative phase absorption. Also, I added x- and y-axis labels to the graph.

5. Regarding the Fig. S4; why you are not trying to remove the lines inside the figure? is odd for me; please clear the graphs and set a clean white background for all of them

Author:

Thanks for the comment. The lines inside the chart were removed.

6. Regarding Fig. S5; please mark the states with significant changes, i cannot follow your text with the figure changes.

Author:

We reviewed the photos again and based on your comment, we decided to remove this picture and section.

7. Why there is no TEM, DLS and FESEM of the used DNC? how can i be sure that your used DNC is in the nano-dimension. You should provide all of the TEM, FESEM and DLS results to prove the nano-sized curcumin.

Author:

Thank you for your strong comment, I added these information in the supplementary. 

(A): DLS diagram of dendrosomal nanocurcumin (DNC) in which was analyzed in terms of size distribution by intensity of light scattering. With regard to DLS measurements (hydrodynamic radius), dendrosome particles size are majorly around 100 nm. 

(B): The size, ζ-potential and polydispersity index (PDI) of DNC.

(C): Transmission electron micrographs (TEM) of DNC: spherical shape of DNC which indicate the compact form of micelles-carriers.

8. The quality of all of the images are really poor; please replace all of the images with high-quality one.

Author:

We have raised the quality of all photos.

9. Please compare your results with the literature; it would be better to add a table regarding this

Author:

Thanks for your good suggestion. We prepared the table and placed the colored columns of our results.

Concentration / Dose

Effect

Reference

Curcumin

0.06-1 

Proliferation/Migration

45,16

LLLT

red 620 -770nm0.3 J/cm2 

, blue 400- 480nm0.3 J/cm2 

, green 470-550nm/0.3 J/cm2 

Proliferation/ Migration

46,47

DNC+LLLT

0.5, 0.75µM + blue 450 nm/0.63, 0.95 j/cm2

Proliferation/Migration

Curcumin

0.3% on day 3

Increase in TGF-β 

52

LLLT

2 J/cm2

Increase in TGF-β 

31

DNC+LLLT

0.75µM + blue 450 nm 0.95 j/cm2

Increase in TGF-β 

Curcumin

Increase in VEGF

36,55

LLLT

red 620 -770nm0.3 J/cm2 

, blue 400- 480nm0.3 J/cm2 

, green 470-550nm/0.3 J/cm2 

Increase in VEGF

46

DNC+LLLT

0.75µM + blue 450 nm 0.95 j/cm2

Increase in VEGF

Curcumin

700 μg/well for 1-3 days

Decrease in TNF-α & IL-6

59,61

LLLT

blue 450 nm 25, 50 , 100 mJ/cm2

Decrease in TNF-α & IL-6

60

DNC+LLLT

0.5, 0.75µM + blue 450 nm 0.95 j/cm2

Decrease in TNF-α & IL-6

Curcumin

Anti-oxidant doses

ROS scavenging

16, 65

LLLT

Blue 456 nm 5 W/m2 

ROS generation

60

DNC+LLLT

0.5, 0.75 µM + blue 450 nm 0.95 j/cm2 

No change in ROS 

10. Also, i suggest to provide the raw data including the flowcytometry raw datas as well as the raw MTT results in the supplementary information; i need to be sure about the validity of your results

Author:

The comment considered, I attached them.

 11. Please be careful about the syntax errors: 450nm is wrong!! 450 nm is in the appropriate form.

Author:

The comment considered, I corrected them.

Dear reviewer2

First, we would like to thank you for reviewing our manuscript. We also thank the reviewers for their careful reading and constructive suggestions. Reviewed manuscript was studied and revised according to reviewers’ comments. The comments were responded point by point and highlighted through the body of manuscript (in this file). It is our belief that the manuscript was substantially improved after applying the valuable reviewers’ comments

Reviewer #2: This is an interesting study that looks at the proliferation, motility of embryonic fibroblasts and the expression profiles TGF-β,VEGF , TNF-α and IL-6 following application of nanocurcumin and 450 nm light.

Author:

We appreciate the very positive and constructive comments from the reviewer. According to your comment, text was corrected and highlighted.

 Before publication, please consider the following comments. P 4 line 5: what culture medium was used?

Author:

The DMEM medium was used. The comment considered, I added and highlighted.

 P4 4. Line 6: Although citations were included, some brief details of this optimized protocol should be included.

Author:

In Brief, different weight/weight ratios of dendrosome/curcumin were analyzed leading to the establishment of an appropriate ratio of 25:1. The loading of curcumin onto DNC was performed in which curcumin and dendrosome were co-dissolved in 5 mL of acetone followed by adding 5 mL of PBS, while stirring constantly. Acetone was evaporated by a rotary evaporator. The curcumin/dendrosome micelle solution was sterilized using a 0.22 μ syringe filter (Millex- LG, Millipore Co., USA). At the end, the prepared DNC was stored at 4 C in a light protected condition until use.

 P4 Line 8: please add the exact model used and/or emission spectrum

Author:

The light source was a diode laser device provided by the RAADMED company from Iran at the wavelength of 450 nm with the output power of 75mW. The radiation spectrum of the diode source (SHARP GH04580A2G) is in the range of 440-460 nm (centred at 450 nm). The distance of sample from the light source was ﬁxed at 6cm with the beam area of light as 6cm2. MEFs were irradiated one time after passing 24 hours from seeding in a 96-well culture plate (the diameter of 7mm) for MTT assay) and within a petri dish with a diameter of 35mm for the rest of experiments. The irradiation was performed in a dark room. The irradiation time was automatically adjusted by the device just by setting the energy (Joule) of the radiation due to applying a constant power by the laser (Energy= Power x Time). Therefore, the cells were irradiated for 224 seconds for getting the energy of 17.9 Joule (with energy density of 0.63 J/cm2) and 337 seconds for the dose of 26.9 Joule (energy density of 0.95 J/cm2). For other doses, the time was set in the same way. 

 P4 line 34: please defined DMEM when first used.

Author:

The comment considered, I added and highlighted. P5. Line 7: paint? Do you mean the pigment?

Author:

We changed this word to colour. 

The intensity of the colour produced after dissolving the deposition of formazan in organic solvents such as dimethyl sulfoxide is measured spectrophoto-metrically.

 P5 Line 10: indeed, the MTT assay is a measure of metabolic activity. Metabolism is highly dynamic so I am unsure one could claim that it is strictly associated with viability?

Author:

Thank you for your informative comment. We added (The reduction of MTT can only occur in metabolically active cells, so this assay represents the level of activity the cells) in the MTT section in the text. And we did cell cycle assay so we added (Since the aforementioned results were in line with the cell cycle assay results, the outcome was considered as the proliferation-inducing effect.) for proliferation. 

 P5 Line 12: what were the doses of nanocurcumin and/or light?

Author:

nanocurcumin (0.25-10 µM) and laser (0.31-25.47 j/cm2) alone and in combination.

 P5. Line 20: which experimental groups? please be specific.

Author:

DNC alone (0.5, 0.75 μM), LLLT 450 nm alone (0.07, 0.10, 0.63 and 0.95 j/cm2 ) and the combination group (DNC (0.5, 0.75 μM) + LLLT 450nm (0.95 j/cm2) P6. Line 4: were the proteins not extracted? Or was it just the supernatant? I would think it would be difficult to get enough protein concentration this way.

Author:

We used supernatant. We did Bradford assay for protein concentration. I added the result below. The Protein concentration was enough. 

 P6 Line 18: the authors need to provide the exact concentrations/doses used.

Author:

The comment considered, I added and highlighted. P 7: Line 13. Were the experiments performed on 3 different days?

Author:

The experiments were performed on 3 different days.

---

## [Decision Letter · Decision Letter 1]

2 Feb 2021

Combination treatment of dendrosomal nanocurcumin and low-level laser therapy develops proliferation and migration of mouse embryonic fibroblasts and alter TGF-β, VEGF , TNF-α and IL-6 expressions involved in wound healing process

PONE-D-20-33956R1

Dear Dr. sadeghizadeh,

We’re pleased to inform you that your manuscript has been judged scientifically suitable for publication and will be formally accepted for publication once it meets all outstanding technical requirements.

Kind regards,

Michael R Hamblin

Academic Editor

PLOS ONE

Additional Editor Comments (optional):

Reviewers' comments:

Reviewer's Responses to Questions

**Comments to the Author**

1. If the authors have adequately addressed your comments raised in a previous round of review and you feel that this manuscript is now acceptable for publication, you may indicate that here to bypass the “Comments to the Author” section, enter your conflict of interest statement in the “Confidential to Editor” section, and submit your "Accept" recommendation.

Reviewer #1: (No Response)

Reviewer #2: All comments have been addressed

2. Is the manuscript technically sound, and do the data support the conclusions?

Reviewer #1: (No Response)

Reviewer #2: Yes

3. Has the statistical analysis been performed appropriately and rigorously? 

Reviewer #1: (No Response)

Reviewer #2: No

4. Have the authors made all data underlying the findings in their manuscript fully available?

Reviewer #1: (No Response)

Reviewer #2: No

5. Is the manuscript presented in an intelligible fashion and written in standard English?

Reviewer #1: (No Response)

Reviewer #2: Yes

6. Review Comments to the Author

Reviewer #1: (No Response)

Reviewer #2: The authors have appropriately responded to my concerns and I recommend acceptance of the manuscript for publication.

7. PLOS authors have the option to publish the peer review history of their article (what does this mean?). If published, this will include your full peer review and any attached files.

Reviewer #1: No

Reviewer #2: No

---

## [Editor Report · Acceptance letter]

8 Feb 2021

PONE-D-20-33956R1 

Combination treatment of dendrosomal nanocurcumin and low-level laser therapy develops proliferation and migration of mouse embryonic fibroblasts and alter TGF-β, VEGF, TNF-α and IL-6 expressions involved in wound healing process 

Dear Dr. sadeghizadeh:

I'm pleased to inform you that your manuscript has been deemed suitable for publication in PLOS ONE. Congratulations! Your manuscript is now with our production department. 

Kind regards, 

on behalf of

Dr. Michael R Hamblin 

Academic Editor

PLOS ONE